# Reduction of Water Color in a Spinning Disc Reactor

Eugenia Teodora Iacob-Tudose [1] , Carmen Zaharia [2,]* and Nicoleta Melniciuc-Puica [3]

1   Department of Chemical Engineering, Faculty of Chemical Engineering and Environmental Protection, "Gheorghe Asachi" Technical University of Iasi, 73 Prof. Dr. Docent Dimitrie Mangeron Blvd., 700050 Iasi, Romania

2   Department of Environmental Engineering and Management, Faculty of Chemical Engineering and Environmental Protection, "Gheorghe Asachi" Technical University of Iasi, 73 Prof. Dr. Docent Dimitrie Mangeron Blvd., 700050 Iasi, Romania

3   Department of Restoration and Preservation of Religious and Cultural Patrimony, Faculty of Orthodox Theology, "Alexandru Ioan Cuza" University of Iasi, 11 Carol I Blvd., 700506 Iasi, Romania

*   Correspondence: czah@ch.tuiasi.ro or carmen.zaharia@academic.tuiasi.ro

**Abstract:** In this study, spinning disc (SD) technology was successfully applied to a synthetic water to remove its color. The preliminary data performed in a regular mixing system using a potential adsorptive material, i.e., double-layered hydroxide of a ZnAlLDH type, did not provide a significant decrease (no more than 10–15%) in the water color content. Thus, ZnAlLDH (2 g/L) was added to the synthetic water containing 50 mg/L Rosso Remazol RB dye that was subsequently fed onto the spinning disc. The SD efficiency was investigated at four different water-supplying flow rates (5.76, 6.00, 7.44 and 8.16 L/h) and four different disc rotational speeds (100, 250, 500 and 800 rpm). The best color removals of 44.39%, 41.14% and 42.70% were obtained at 6 L/h and 250 rpm, 6 L/h and 500 rpm and 5.76 L/min and 800 rpm, respectively, in only a 50 min working time period. In addition, for a relatively low color concentration in water (~30 mg/L dye) and at the lowest electric power consumption, Fenton oxidation was performed in the SD setup for a more advanced color removal of 62.54% within a 50 min time period. Furthermore, two other materials, titanium and aluminium oxides, underwent similar investigations in the SDR setup, and the obtained results were comparatively discussed. The FTIR spectra of each solid material before and after the SD technology application were used to appreciate the dye-retention performance of each material used. The obtained results indicated that the spinning disc technology correlated with the tested materials could significantly improve the water color (over 40% color reduction), this level of color reduction being higher than that obtained following a coagulation–flocculation test (20–28% color reduction), an ion exchange (25–30% color removal) or a sand filtration step (15–20%) applied to the same dye-based water sample. A further increase in color removal could be achieved by using an additional oxidative step (more than 65% color reduction).

**Keywords:** color removal; ZnAlLDH; titanium and aluminium oxides; spinning disc reactor; Fenton oxidation; adsorption efficiency

## 1. Introduction

Color is an indicator of general water quality that can significantly influence the quality of the final technological effluent, or the natural and artificial processes occurring in freshwater resources and also the technological processes supplied with it. Water color can be due to different organic and inorganic species present in small or high amounts in solid or dissolved forms, such as dyes, aquatic plants or other colored products among others. Synthetic reactive dyes (residual ones) found in different potential water resources are usually known to be persistent organic micropollutants, which can be immediately visualized by a change and/or intensification of water color, and their presence can be eliminated, or partially reduced, by different water treatments. Numerous practices based on

ion exchange (with synthetic ionic exchangers), advanced chemical oxidation, biodegradation, electrocoagulation, ultrafiltration or other membrane processes and adsorption [1–4], among others, are used to remove residual dyes from water resources. Adsorption is one of the most useful techniques in the removal of reactive dyes from different effluents/water resources due to its numerous advantages such as its simple design, easy operation and the relatively simple regeneration of the synthetic adsorptive materials utilized, but its use in association with SD technology has not been intensively studied and applied.

Previous experimental studies performed by the authors on real textile wastewaters containing 2% w. bentonite, within a rotating disc setup (SD), indicated good removal rates from 50% up to 60% for suspended solids and turbidity, within short time periods of 25 min, for different investigated values of effluent flow rate and disc rotational speed without any additional treatment [5,6]. However, the color removal obtained in the same experimental setup and working conditions using synthetic-colored water samples revealed smaller values for most of the investigated operational parameters/variables within the applied SD technology. Thus, the main purpose of this study was to investigate the color removal efficiency within a spinning disc setup using different potential adsorptive materials added to the effluent (synthetic-colored water containing a reactive dye) prior to being fed to the disc. In addition, comparisons between regular mixing systems containing different materials tested as potential adsorbents and their use in a rotating disc setup were performed to assess whether the spinning disc technology improved the color removal from aqueous effluents. The selected materials chosen for water treatment in the SD setup were the double-layered hydroxide of a ZnAlLDH type, and also titanium oxide ($TiO_2$) and aluminum oxide ($Al_2O_3$). These substances are known to display good photocatalytic activity [7–9] as a result of interplay among different influencing factors such as particle size and mobility, surface area, degree of aggregation, phase composition, adsorption of molecules from aqueous phase, interactions between adsorbed species and the nature of the solvent [10–22]. However, in this study, only the adsorption performance of the above-mentioned selected materials was tested within the spinning disc laboratory experimental setup, for four different water flow rates and four different rotating disc speeds. The dye adsorption onto the tested adsorptive materials was highlighted by their presence in the FTIR spectra of specific peaks originating from the characteristic groups of the studied reactive dye. Additionally, for the studied double-layered hydroxide (ZnAlLDH) as an adsorptive material, at the end of the measurements, Fenton oxidation was applied in the SD system to control whether a higher removal of color degrees was attained. The association of adsorption with advanced oxidation (i.e., Fenton oxidation) seemed to be beneficial for increasing the water color reduction.

## 2. Materials and Methods

*Adsorbate.* The aqueous stock solution of Rosso Remazol BR (RR) dye (600 mg/L dye), prepared on the day of use, was added to a 10 L tank containing distilled water in order to attain an initial dye concentration of around 50 mg/L. The RR dye was a commercial reactive azo dye characterized by a molecular weight (MW) of 824 g/mol, a maximum absorption ($\lambda_{max}$) at 517 nm, the chemical formula of $C_{20}H_{12}N_6O_{16}S_5Na_4$ and a purity of 82.60%. It is used to print textile fabrics based on cotton or wool for a final red and/or orange color of the final product/fabric. No other additives or/and adjuvants were added in the prepared aqueous solution, which was considered as a basic reference/model of synthetic-colored water.

*Adsorbents.* The tested material was added into the tank at the concentrations specified in Table 1.

**Table 1.** Material concentrations in the colored water fed to the spinning disc.

| Material Type | ZnAlLDH | TiO$_2$ | Al$_2$O$_3$ |
|---|---|---|---|
| Concentration (w) % | 20 | 20 | 20 |

The tested ZnAlLHD was a nano-structured material of double-layered hydroxides (LDHs) and was a Zn- and Al-substituted hydrotalcite-like anionic clay, which is obtained by the co-precipitation method at a constant pH and with vigorous stirring [19–21]. Aqueous solutions of $Zn(NO_3)_2 \cdot 6H_2O/Al(NO_3)_3 \cdot 9H_2O$ were used together with aqueous solutions of NaOH and $Na_2CO_3$ added dropwise together to maintain a constant pH. The resulting precipitates were aged (5 h), separated by centrifugation, washed and dried in an oven at 338 K.

The commercial $TiO_2$ (white) purchased from Chemicals Co., (Iasi, Romania), used especially in the construction industry, with a specific surface area of ~200 $m^2/g$ and particle sizes lower than 0.5 mm, was tested.

Activated alumina was purchased as a commercial product from Chemicals Co., (Iasi, Romania), it being a highly porous material with a surface area in the range of 30–115 $m^2/g$ and very low particle sizes (<0.5 mm).

*Other chemicals.* High-analytical-purity reagents were used for Fenton oxidation in the SD system, especially for the preparation of the aqueous solutions of $FeSO_4$ (6 g/L, solid $FeSO_4 \cdot 7H_2O$ from Chemical Co., Iasi, Romania), $H_2O_2$ (30%, Nordic Invest SRL Co., Cluj Napoca, Romania) and $H_2SO_4$ (concentrated, Merck Co., Kenilworth, NJ, USA and 1N, Chemical Co., Iasi, Romania).

The water pH was directly measured with a Combo portable pH/EC/TDS Tester (Hanna Instruments Inc., Woonsocket, RI, USA) and the water absorbance was measured using a DR/2000 Direct Reading spectrophotometer (Hach Co., Bucuresti, Romania).

The experimental setup used in this study comprised a 20 cm diameter rotating disc encased in a plexiglass rectangular box as described in a previously published paper [5]. The synthetic-colored water was fed onto the disc from a storage tank using a centrifugal pump. Additional constructive characteristics of the experimental setup can be found in a previous paper [5]. The colored water temperature was continuously monitored and found to be $30 \pm 5\,°C$. The samples were collected right at the exit from the disc and their absorbance ($A_{456}$) for color determination was measured with a DR/2000 spectrophotometer, since the water color analysis was based on the absorbance measurement at 456 nm and its conversion in Hazen color units (i.e., an absorbance of 0.056 corresponds to 50 Hazen color units (HU)), or only its absorbance measurement at three specific wavelengths (430, 525 and 630 nm) [4–6].

The validation of possible RR dye adsorption onto the tested adsorbent (based on the identification of the functional groups existing in the original adsorbent, which could be involved in the binding of dye molecules) was controlled by Infrared Fourier Transform (FTIR) spectroscopy. The FTIR spectra were drawn comparatively for the dye and the adsorbent before and after the treatment in the experimental SD setup to highlight how the molecules binded to the adsorbent but also to investigate possible changes in the initial structure of the adsorbent as a result of the dye adsorption. Thus, the FTIR spectra were recorded using a Vertex 70, Bruker FTIR spectrophotometer (equipped with an EasiDiff diffuse reflectance sampling accessory (Pike Technologies, Fitchburg, WI, USA) and a data processor with Spectra Manager and an internal standard normalization of spectra) in the diffuse reflectance mode (DR FTIR) in the wavelength range 4000–250 $cm^{-1}$ with a spectral resolution of 4 $cm^{-1}$ and 16 scans, at room temperature and a humidity of 10%. All samples were ground with spectrophotometric KBr in an agate mortar (KBr powder was used as background of the spectrum).

Based on the experimental data obtained from the SD setup, the removal efficiency (%), R, was determined using Equation (1):

$$R\ [\%] = [(C_i - C_f)/C_i] \times 100 \tag{1}$$

where, $C_i$ is the initial value of the color indicator (absorbance, or HU) and $C_f$ is the measured value of the color indicator (absorbance, or HU) at a specific time, t.

## 3. Results

### 3.1. Colored Water Treatment with Selected Adsorptive Materials in SD Experimental Setup

The use of solid ZnAlLDH material as an adsorbent in a regular mixing system to diminish the Rosso Remazol (RR)-based color content of the different samples of synthetic-colored water indicated increases in the sample absorbances. The investigated samples of 6 mg/L, 12 mg/L and 18 mg/L RR dye concentrations, respectively, were mixed for two hours in media of different pH values, as presented in Table 2, which includes the measured absorbance values, every 30 min.

**Table 2.** Measured absorbances ($A_{456}$) of three different colored water samples containing 0.1 g of ZnAlLDH/50 mL (2 g/L) at different mixing times.

| Investigated Samples | t = 0 min | t = 30 min | t = 60 min | t = 90 min | t = 120 min |
|---|---|---|---|---|---|
| Sample 1 (pH = 7) | 0.344 | 0.182 | 0.233 | 1.292 | 2.003 |
| Sample 2 (pH = 2.5) | 0.330 | 1.429 | 1.758 | 2.051 | 2.071 |
| Sample 3 (pH = 4.5) | 0.374 | 0.305 | 0.659 | 1.052 | 1.340 |

The first and third samples, corresponding to an initial water pH of 7 and 4.5, respectively, displayed a decrease in the absorbance value after the first 30 min, meaning an initial color decrease (due to dye adsorption onto the tested adsorbents), followed by continuous increases in their absorbance values, which could be due to the mixing regime being favorable to the destabilization of suspended solids (great solid dispersion and increased turbidity) or the desorption of adsorbed dye from the external surface of solid adsorbents in the water sample due to solid mass transport (the action of the resultant force between centrifugal and gravity forces). Furthermore, all the analyzed samples exhibited increases in the measured absorbance over time, indicating the poor adsorption qualities of the ZnAlLDH for the studied RR dye, or the size variation in the solids in water. Therefore, the operation times in the future SD testing for water discoloration were selected to be no higher than 60 min without use of recycling facilities and with no high electricity consumption.

Previous discoloration treatment using bentonite, performed within a spinning disc system for real textile wastewater [5], indicated reasonable decreasing values of sample absorbance and, thus, an increased dye removal efficiency of the SD technology. Thus, a similar treatment applied in the SD system was performed to verify its efficiency in the synthetic-colored water treatment, where solid ZnAlLDH was used as an adsorbent, among others.

The last two materials (titanium and aluminium oxides) were also tested within the experimental SD setup to confirm their adsorbent availability. For data comparison, only the results performed in neutral pH conditions of the prepared synthetic-colored water, where the best results for ZnAlLDH as adsorbent were obtained, are presented.

### 3.1.1. Influence of Colored Water Flow Rate on Discoloration for All Tested Adsorbents

*Testing ZnAlLDH as an adsorbent in the SD setup.* The influence of the colored water flow rate treated with ZnAlLDH (2 g/L) at four different constant spinning disc rotational speeds, namely 100, 250, 500 and 800 rpm, respectively, is indicated in Figure 1a–d, where the abscissa indicates the processing residence time.

For all the investigated flow rate values of 5.76, 6.00, 7.44 and 8.16 L/h, the color removal increased, attaining the highest measured value of 49.03% after a 55 min working time period at 250 rpm. Other operating parameter values rendering data higher than 40%, which is useful for water discoloration in an SD setup, as presented in Figure 1, are synthesized in Table 3.

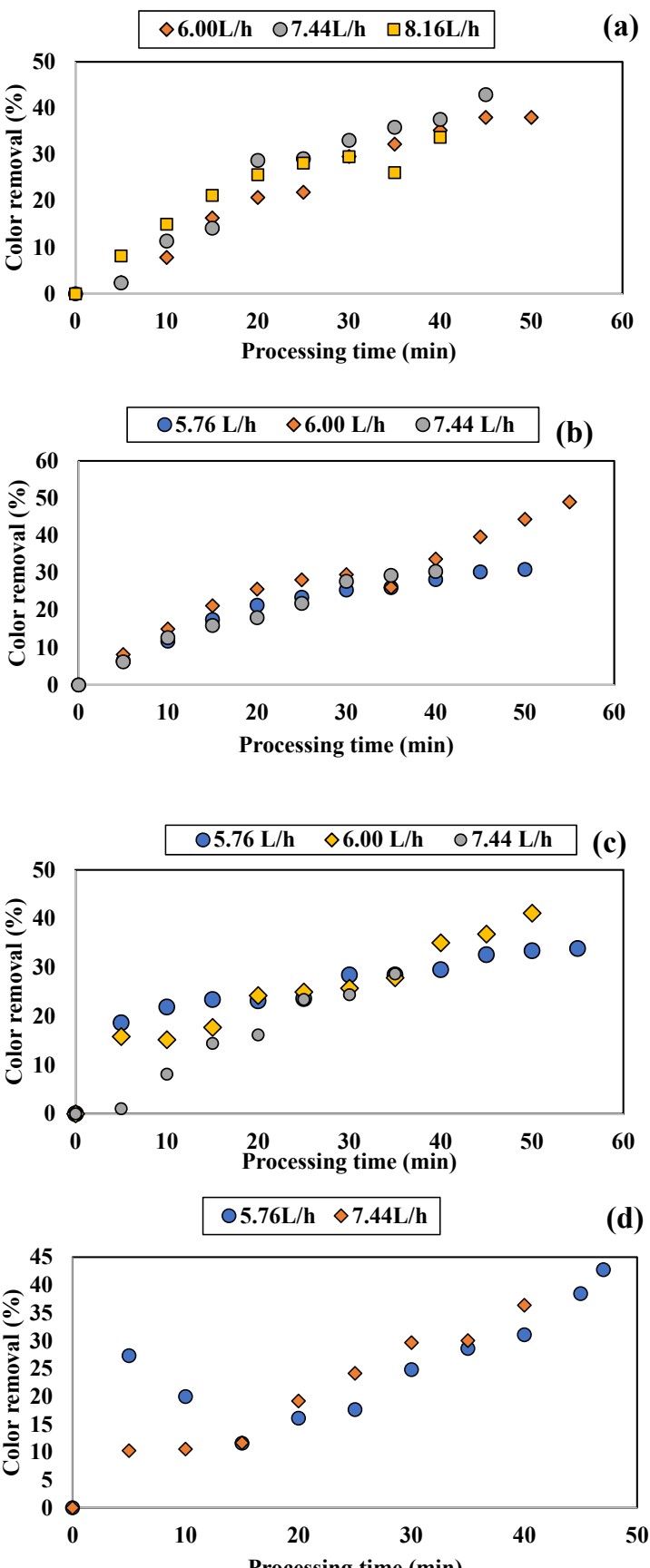

**Figure 1.** Colored water flow rate influence on RR-dye-based color removal in an SD system with ZnAlLDH: (**a**) 100 rpm, (**b**) 250 rpm, (**c**) 500 rpm, (**d**) 800 rpm.

**Table 3.** Maximum discoloration values for different colored water flow rates and disc rotational speeds using ZnAlLDH within an SD experimental setup.

| Rotation Speed [rpm] | Flow Rate [L/h] | Color Removal [%] | Time Period [min] |
|---|---|---|---|
| 100 | 6.00 | 38.01 | 60 |
|  | **7.44** | **42.92** | 45 |
|  | 8.16 | 40.00 | 30 |
| 250 | 5.76 | 30.94 | 50 |
|  | **6.00** | **49.03** | 55 |
|  | 7.44 | 30.39 | 40 |
| 500 | 5.76 | 33.87 | 55 |
|  | **6.00** | **41.14** | 50 |
|  | 7.44 | 28.68 | 35 |
| 800 | **5.76** | **42.74** | 47 |
|  | 7.44 | 36.37 | 40 |

Table 3 includes the maximum discoloration values at different disc rotational speeds and water flow rates attained within the SD experimental setup. One can observe that these values ranged between ~30% and almost 50% color removal in the SD setup, within a less than one hour working time period. The maximum obtained color removal and the corresponding working conditions are highlighted in bold font.

*Testing TiO$_2$ as an adsorbent in the SD setup.* The colored water treated with solid TiO$_2$ (2 g/L) at the same colored water flow rate values of 5.76, 6.00, 7.44 and 8.16 L/h, respectively, at constant disc rotational speeds of 100, 250, 500 and 800 rpm, rendered increasing trends of color removal over time, with the maximum attained values ranging from ~40% to the highest of 67.55% (obtained at 6.00 L/h and 800 rpm, after 70 min), as presented in Figure 2a–d. Based on the obtained results, TiO$_2$ was confirmed as having good adsorbent properties when used in correlation with the SD technology. A somewhat similar system formed of TiO$_2$ stored as a thin layer on a spinning disc was used to investigate the photo-oxidation of Methylene Blue dye dissolved in a water stream, with the largest reaction rate reported at 100 and 200 rpm [7].

The maximum color removal values obtained using TiO$_2$, in an SD experimental setup together with the corresponding working conditions, namely water flow rate and disc rotational speed values, are included in Table 4.

**Table 4.** Maximum discoloration values for tested water flow rates and disc rotational speeds using TiO$_2$ within an SD experimental setup.

| Rotation Speed [rpm] | Flow Rate [L/h] | Color Removal [%] | Time Period [min] |
|---|---|---|---|
| 100 | 5.76 | 52.74 | 95 |
|  | 6.00 | 52.84 | 90 |
|  | 7.44 | 45.98 | 59 |
|  | **8.16** | **56.52** | 55 |
| 250 | **5.76** | **61.00** | 90 |
|  | 6.00 | 52.85 | 84 |
|  | 7.44 | 46.62 | 60 |
| 500 | **5.76** | **56.90** | 85 |
|  | 6.00 | 46.93 | 75 |
|  | 7.44 | 56.08 | 95 |
| 800 | 5.76 | 39.23 | 90 |
|  | **6.00** | **67.55** | 70 |
|  | 7.44 | 59.68 | 93 |

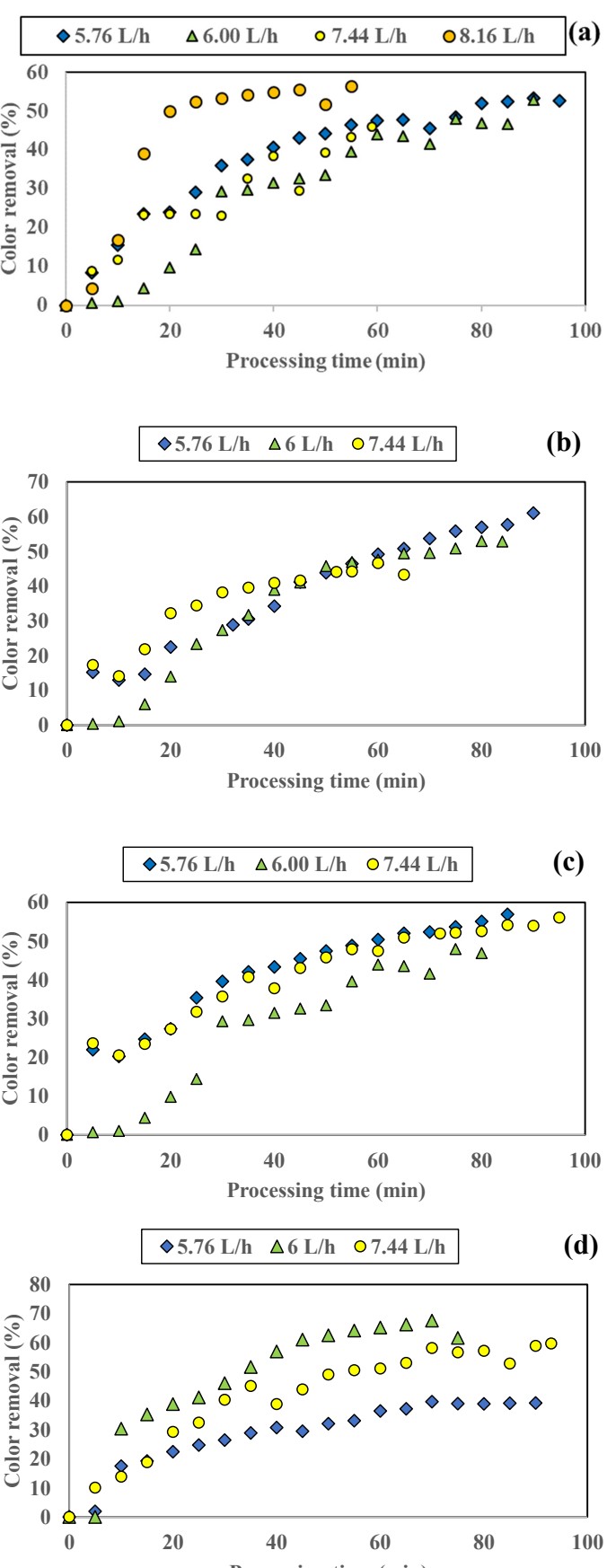

**Figure 2.** Water flow rate influence on RR-dye-based color removal in an SD system with TiO$_2$ at constant disc rotational speeds: (**a**) 100 rpm, (**b**) 250 rpm, (**c**) 500 rpm, (**d**) 800 rpm.

*Testing Al$_2$O$_3$ as an adsorbent in the SD setup.* For all the above-mentioned water flow rates and disc rotational speeds, the colored water treated with solid Al$_2$O$_3$ used as an adsorbent and collected from the spinning disc displayed fluctuating absorbance values over time, indicating poor adsorbent properties for the RR-dye-containing water discoloration. All the obtained data are included in the Supplementary Materials, Table S1.

### 3.1.2. Influence of Disc Rotational Speed on Discoloration for All Tested Adsorbents

*Testing ZnAlLDH as an adsorbent in the SD setup.* The influence of spinning disc speed when RR-dye-based colored water was treated with ZnAlLDH (2 g/L) at four different constant water flow rates of 5.76, 6.00, 7.44 and 8.16 L/h is indicated in Figure 3a–c. As already mentioned above, the highest color removal, in this case, was recorded at 250 rpm and after a 55 min working time period and a low water flow rate (6 L/h).

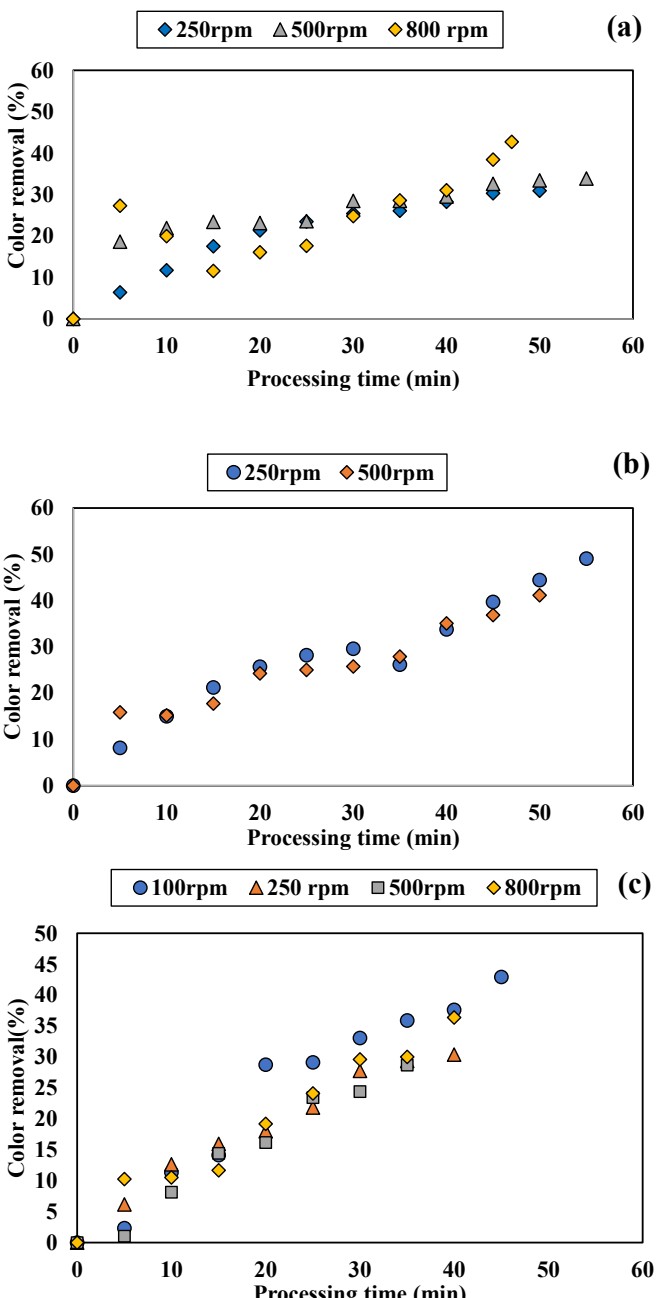

**Figure 3.** Disc rotational speed influence on RR-dye-based water color removal in an SD system with ZnAlLDH at constant water flow rates: (**a**) 5.76 L/h, (**b**) 6.00 L/h, (**c**) 7.44 L/h.

*Testing TiO$_2$ as an adsorbent in the SD setup.* The colored water absorbances measured when using TiO$_2$ to remove RR-dye-based water color at different spinning disc speeds, for different constant water flow rates, is presented in Figure 4, with the largest color removal value registered at 800 rpm.

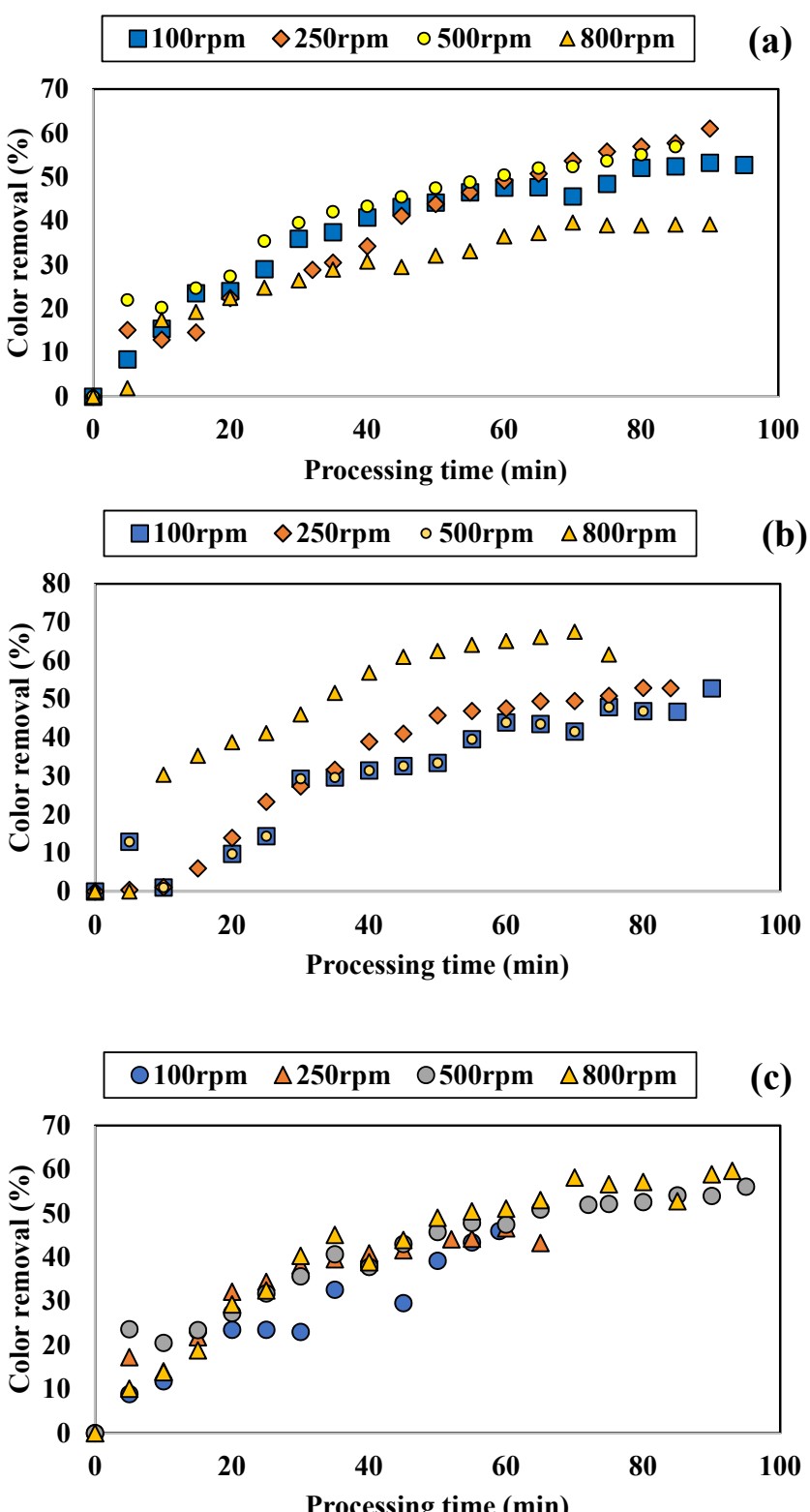

**Figure 4.** Disc rotational speed influence on RR-dye-based water color removal in an SD system with TiO$_2$ at constant liquid flow rates: (**a**) 5.76 L/h, (**b**) 6.00 L/h, (**c**) 7.44 L/h.

### 3.1.3. Application of Additional Fenton Oxidation on Colored Water Tested with ZnAlLDH in the SD Setup

In the SD experimental setup, the Fenton oxidation (FO) process in acidic conditions was used on the colored water utilizing different amounts of $H_2O_2$ (30%) activated by $FeSO_4$ (stock solution of 0.34 mmol/L) and $H_2SO_4$, which produces hydroxyl radicals (HO·), which are known to have stronger oxidative properties than $H_2O_2$. To ensure an acidic pH, $H_2SO_4$ (concentrated and 1N solution) was added. The FO was applied in the working conditions corresponding to the highest color removal values for ZnAlLDH, at 6 L/h and 250 rpm, and for $TiO_2$ in the low electric power consumption conditions chosen both for the transport pump and for the motor that drives the disc (i.e., a low water flow rate of 6.00 L/h and disc rotational speed of 250 rpm). The data registered for the two different adsorptive materials in the SD setup in association with FO are presented in Figures 5 and 6, respectively, in comparison to the experimental data obtained in the simple SD configuration at the same working conditions (no FO applied). For ZnAlLDH, Figure 5 indicates an increase in the water discoloration when FO was used in the SD configuration, with the maximum value of 62.54% color removal being obtained after a 50 min working time period. For $TiO_2$, the data for the colored water treatment with FO in the SD system indicated a higher discoloration than for the simple SD process in the first 35 min of the working time period. Afterwards, similar discoloration values seemed to be obtained for both the FO in the SD and simple SD systems.

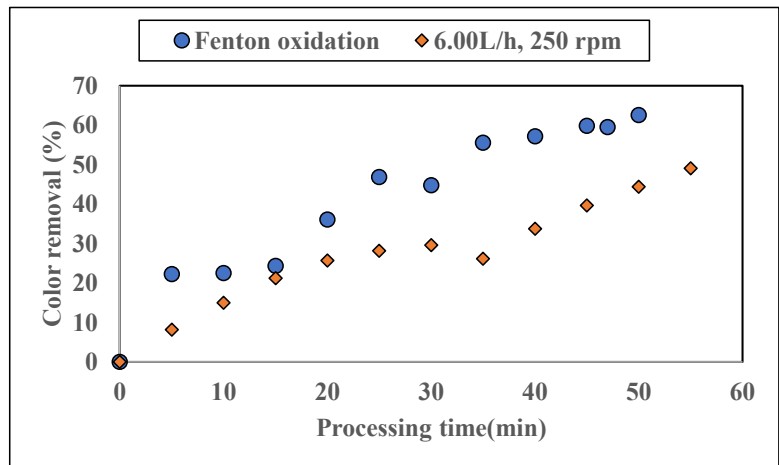

**Figure 5.** Comparison of color removal for FO in SD and SD systems at 6 L/h water flow rate and 250 rpm with ZnAlLDH as adsorbent.

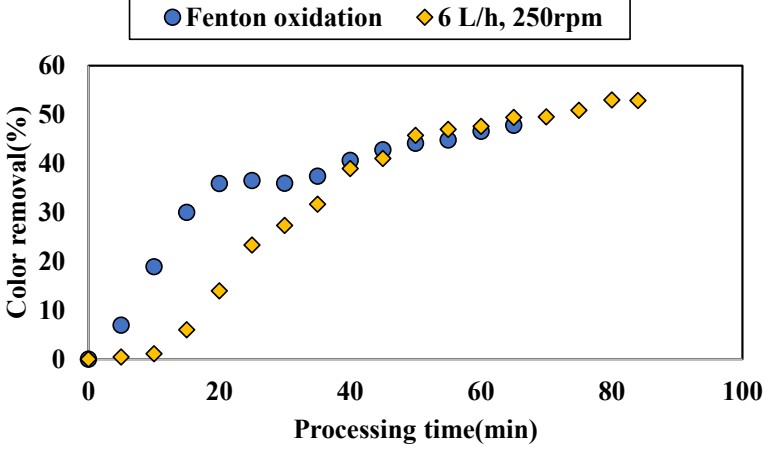

**Figure 6.** Comparison of color removal for FO in SD and SD systems at 6 L/h water flow rate and 250 rpm with $TiO_2$ as adsorbent.

### 3.2. Validation of Dye Retention onto Selected Materials by FTIR Spectroscopy

The FTIR spectra in the diffuse reflectance mode obtained for the three tested adsorbents (ZnAlLDH, TiO$_2$ and Al$_2$O$_3$) before and after the dye adsorption, as well as the spectrum of the Rosso Remazol RB dye, are presented in Figure 7a–c.

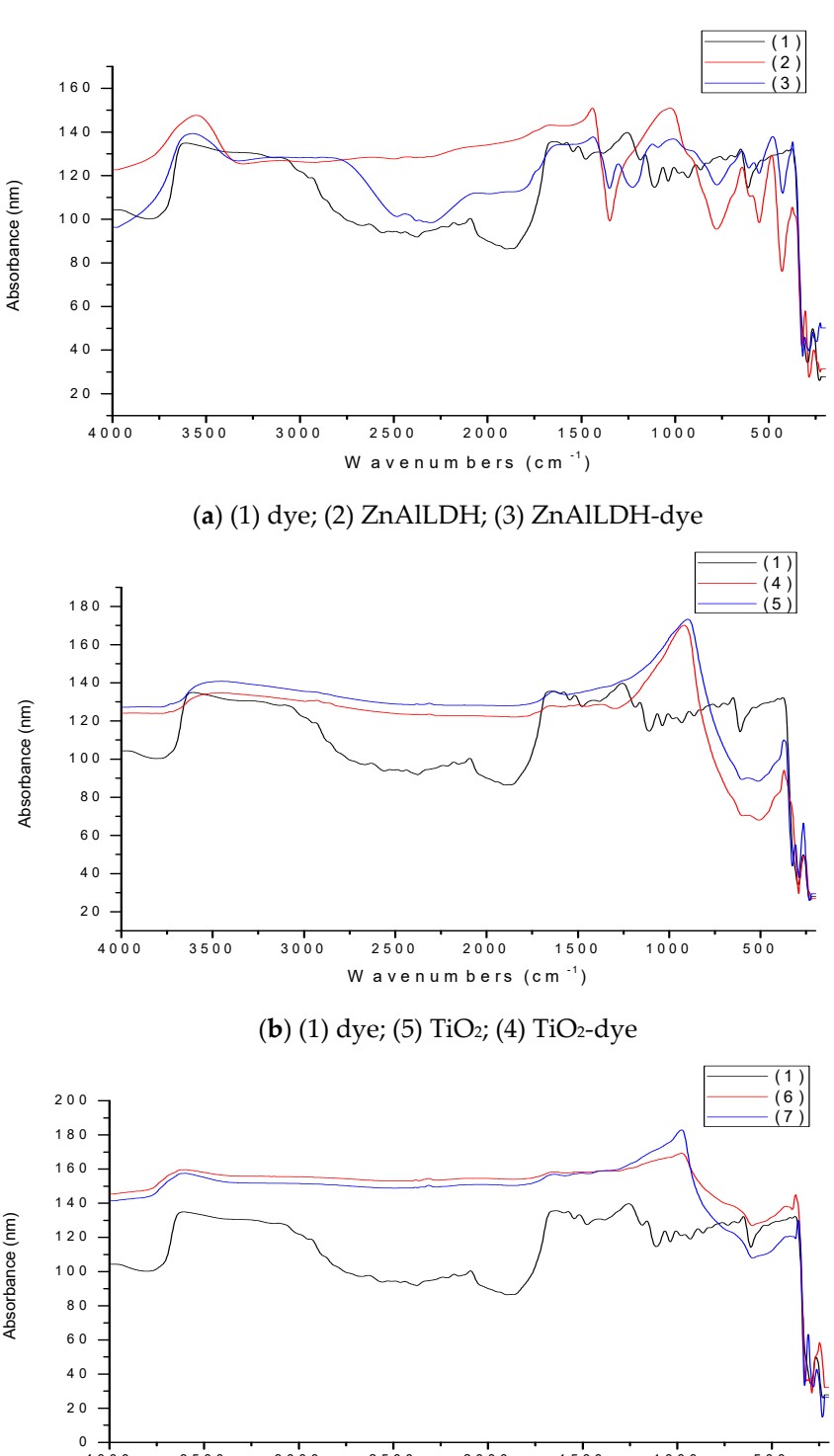

(**a**) (1) dye; (2) ZnAlLDH; (3) ZnAlLDH-dye

(**b**) (1) dye; (5) TiO$_2$; (4) TiO$_2$-dye

(**c**) (1) dye; (6) Al$_2$O$_3$; (7) Al$_2$O$_3$-dye

**Figure 7.** FTIR spectra of RR dye adsorbent before and after dye treatment in SD experimental setup. (**a**) ZnAlLDH; (**b**) TiO$_2$; (**c**) Al$_2$O$_3$.



The analysis of the spectra led to the conclusion that new peaks appeared in the structure of the adsorbent loaded with dye in comparison to its initial structure. In addition, one can notice a decreased intensity of the peaks due to the adsorption process in the case of ZnAlLDH (Figure 7a), with similar decreases in the case of TiO$_2$ (Figure 7b) and an insignificant adsorption in the case of Al$_2$O$_3$ (Figure 7c).

The relevant peaks and their intensity were determined and are summarized in Table 5.

**Table 5.** Representative peaks in the FTIR spectra.

| Solid Material Tested | SD Treatment Status | Relevant Peaks (cm$^{-1}$)/Its Magnitude (s—Strong; m—Medium; w—Weak) |
|---|---|---|
| RR dye (1) | no | 3608 sh, 2093 s, 1641 sh, 1517 s, 1256 s, 1161 m, 1065 s, 891 w, 650 s, 378 s, 267 s |
| ZnAlLDH | Before (2) | 3555 sh, 1441 s, 1027 s, 642 s, 485 s, 375 s, 306 s, 260 m |
| | After (3) | 3571 sh, 3104 w, 2441 m, 1428 s, 1304 s, 1121 m, 1011 s, 644 s, 581 m, 488 s, 373 s, 309 m, 266 m, 226 m |
| TiO$_2$ | Before (5) | 3453 sh, 1630 w, 898 s, 373 s, 308 s, 265 s |
| | After (4) | 3454 sh, 1370 w, 919 s, 371 s, 261 s |
| Al$_2$O$_3$ | Before (6) | 621 sh, 977 s, 428 m, 374 s, 301 w, 248 s |
| | After (7) | 3598 sh, 977 s, 360 s, 308 s, 264 s |

## 4. Discussion

### 4.1. FTIR Spectra of Adsorbents before and after Dye Adsorption in the SD Setup

The FTIR spectra (Figure 7) indicate whether the dye adsorption onto each studied adsorbent took place and also allow for a few remarks on the dye and each adsorbent structure before and after treatment by the SD technology for possible discoloration:

- The peak at 3608 cm$^{-1}$ can be seen in the spectrum of the pure dye (1), it being assigned to N-H in the amine groups due to strong stretching vibrations, since no OH was present in the dye structure. In addition, the peak at 3555 cm$^{-1}$ from ZnAlLDH structure was assigned to strong OH stretching vibrations, and the ones at 3571 and 3104 cm$^{-1}$, from the ZnAlLDH loaded with dye (ZnAlLDH-dye), were assigned to OH stretching vibrations in the LDHs structure or water molecules and to the aromatic benzene nucleus of adsorbed dye, respectively. The peaks at 3453 cm$^{-1}$ from the TiO$_2$ structure, and also at 3454 cm$^{-1}$ from the structure of TiO$_2$ loaded with dye, were assigned to OH stretching vibrations in the TiO$_2$-based adsorbent and water molecules, and the same considerations were made for the peaks at 3621 cm$^{-1}$ from the Al$_2$O$_3$ structure and 3598 cm$^{-1}$ from the Al$_2$O$_3$-dye structure, but no significant dye adsorption inside of the used adsorbent was shown.
- The peak at 2441 cm$^{-1}$ for the ZnAlLDH structure after treatment in the SD setup was assigned to the presence of a carboxylic acid (COOH) originating from the studied dye structure adsorbed onto adsorbent surface.
- The 1630 cm$^{-1}$ peak for the TiO$_2$ structure was assigned to the strong stretching vibrations of C=N double bonds in the aromatic species adsorbed on it.
- The strong deformation vibrations at 1441 and 1428 cm$^{-1}$ from the structure of the ZnAlLDH before and after treatment in the SD experimental setup were assigned either to the C-H deformation vibrations from the dye structure, indicating dye adsorption onto the solid adsorbent surface, or to the aliphatic hydroxyl group deformation due to the adsorbent and water molecules. The peaks at 1304 cm$^{-1}$ from the ZnAlLDH structure after treatment and at 1370 cm$^{-1}$ from the TiO$_2$ structure after treatment in the SD experimental setup could be assigned either to the O=S=O in the sulfonic groups of the studied dye as result of antisymmetric stretching vibrations, or to NO$_2$ deformation vibrations due to the active sites of the dye adsorbed onto the TiO$_2$ sur-

face with the help of water molecules. The 1027 cm$^{-1}$ band for the ZnAlLDH structure before treatment and the 1011 or 1121 cm$^{-1}$ bands from the ZnAlLDH structure after treatment in the SD experimental setup corresponded to the symmetric stretching vibrations of the adsorbent overlapping with other functional groups present in the structure of the adsorbed dye as a sulfonic $SO_3^-$ group. These are valuable indicators that suggest the retention of the dye onto the adsorbent.

- For the region of 1000–250 cm$^{-1}$ (fingerprint), all the peaks corresponded to each characteristic spectrum of tested solid material (with no modifications after treatment), except in the case of the 919 cm$^{-1}$ band for the TiO$_2$ structure, with possible weak dye traces, or of the 644–650 cm$^{-1}$/266–267 cm$^{-1}$ bands for the ZnAlLDH material.

### 4.2. Performance of SD Reactor in Water Color Reduction in the Selected Working Regime

For the RR-dye-based colored water treated with ZnAlLDH in a spinning disc setup, the water flow rate, within the mentioned limit values, could influence to some extent the color removal. However, the disc rotational speed, especially in the low to medium range, seemed to have a stronger influence. It is difficult to explain these types of effects, however the SD efficiency lay within the strong micromixing characteristics, which was enhanced especially at higher disc rotational speeds, and a sufficiently large residence time of the water on the disc was such that the intense mixing benefits manifested in the process occurring on the disc such as water discoloration due to adsorption in connection with the SD technology [7,9]. Furthermore, the size of the tested adsorbent particles influenced the adsorption characteristics and, thus, the color removal efficiency.

Additionally, in comparison to a common batch system using ZnAlLDH as an adsorbent, the SD technology based on the same ZnAlLDH rendered a significant improvement in terms of water discoloration.

Based on Table 4's data obtained for the TiO$_2$-treated colored water, one can conclude that the highest investigated spinning disc speed of 800 rpm combined with an average water flow rate of 6.00 L/h rendered the highest color removal value of 67.55%. In addition, the low flow rates associated with medium spinning disc speeds, similar to the conditions used in case of ZnAlLDH, provided good color removal results, namely 52.85% at 6.00 L/h and 250 rpm operation conditions.

A comparison between the color removal data obtained in the SD setup, when using ZnAlLDH and TiO$_2$, indicated better discoloration values for the latter material for all the investigated colored water flow rates and disc rotational speeds.

Al$_2$O$_3$ as an adsorptive material did not display good adsorbent properties within the SD configuration.

Furthermore, the use of the Fenton oxidation process in the spinning disc system supplied with colored water and ZnAlLDH provided an even better color removal than in the case of the simple SD with the same tested adsorptive material treatment. For the treatment with TiO$_2$, FO applied in the SD system provided a higher discoloration value of 35.92% after only a 20 min working time period, which was higher than the 13.95% color removal obtained in the simple SD system. After further system operation up to 65 min, the color removals obtained in the simple SD and the FO in SD systems seemed to be close enough.

### 5. Conclusions

An experimental study on three different potential adsorptive materials, ZnAlLDH, TiO$_2$ and Al$_2$O$_3$ for water color removal caused by the presence of Rosso Remazol dye in a synthetic-colored water treated in a spinning disc setup was performed. The first two materials rendered good color removal values of 49.03% and 67.55%, respectively, for different working conditions, i.e., 250 rpm and 6 L/h after 55 min, and 800 rpm and 6 L/h after 90 min of the SD reactor working. The best discoloration was obtained when using TiO$_2$ as an adsorbent. The Al$_2$O$_3$ material did not exhibit good adsorptive properties within the SD system. Additionally, Fenton oxidation applied in the SD configuration provided

improved color removal values, especially for the ZnAlLDH-treated water. The FTIR spectra of the raw materials and separated solids after the SD experimental treatment were analyzed to highlight the dye retention onto each adsorptive material used and underlined our findings. SD technology could be applied to reduce water color by adding either solid $TiO_2$ or ZnAlLDH as adsorbents, with over a 32% increase in color reduction without any other additional treatment steps.

**Supplementary Materials:** The following supporting information can be downloaded at: https://www.mdpi.com/article/10.3390/app122010253/s1, Table S1: Absorption ($A_{456}$) values for tested water flow rates and disc rotational speeds using $Al_2O_3$ within an SD experimental setup.

**Author Contributions:** Conceptualization, E.T.I.-T. and C.Z.; methodology, C.Z. and E.T.I.-T.; validation, C.Z. and E.T.I.-T.; formal analysis, C.Z. and E.T.I.-T.; investigation, C.Z., E.T.I.-T. and N.M.-P.; resources, C.Z., E.T.I.-T. and N.M.-P.; data curation, C.Z.; writing—original draft preparation, E.T.I.-T. and C.Z.; writing—review and editing, C.Z. and E.T.I.-T.; visualization, E.T.I.-T., N.M.-P. and C.Z.; supervision, C.Z.; funding acquisition, C.Z. and E.T.I.-T. All authors have read and agreed to the published version of the manuscript.

**Funding:** This research received no external funding.

**Institutional Review Board Statement:** Not applicable.

**Informed Consent Statement:** Not applicable.

**Data Availability Statement:** MDPI Research Data Policies, at https://www.mdpi.com/ethics (accessed on 10 October 2022).

**Conflicts of Interest:** The authors declare no conflict of interest.

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
