# Peer review of "Reduction of Water Color in a Spinning Disc Reactor"

_applsci, doi:10.3390/app122010253_

Round 1

Reviewer 1 Report

Comments in pdf-file

Reviewer 2 Report

Please rewrite and organize the results well because it’s difficult to understand and may cause some confusion. Please consider the importance of figures or tables to support the conclusion and no repetition.

1.     In the abstract, line 27 what does FT-IT means?

2.     Please check the abbreviation of FT-IR or FTIR throughout the manuscript.

3. Section 2. Materials and Methods: please indicate how to obtain material type ZnAlLDH.

4.     Recheck Figure 2 (d) 800 rpm.

5.  This study provides only FTIR analysis before and after dye treatment, in case of studying different types of adsorbents, various characterization techniques may be useful for discussion of the results.

6. Table 4 should be modified for easier understanding.

7.     Figure 6: please add the variable names and their units on the y-axis.

8.    Discussion section should have the subsection related to the section of results.

9.  Conclusion does not provide enough a clear interpretation of the results of this study.

Round 2

Reviewer 2 Report

Authors have modified and corrected to improve their paper so I think that it is possible to publish in this journal.